# Prognostic Accuracy of the qSOFA Score for In-Hospital Mortality in Elderly Patients with Obstructive Acute Pyelonephritis: A Multi-Institutional Study

**DOI:** 10.3390/diagnostics11122277

**Published:** 2021-12-05

**Authors:** Yudai Ishikawa, Hiroshi Fukushima, Hajime Tanaka, Soichiro Yoshida, Minato Yokoyama, Yoh Matsuoka, Yasuyuki Sakai, Yukihiro Otsuka, Ryoji Takazawa, Masataka Yano, Tetsuro Tsukamoto, Tetsuo Okuno, Akira Noro, Katsushi Nagahama, Shigeyoshi Kamata, Yasuhisa Fujii

**Affiliations:** 1Department of Urology, Tokyo Medical and Dental University, 1-5-45 Yushima, Bunkyo-ku, Tokyo 113-8519, Japan; judas1ss1ss@hotmail.co.jp (Y.I.); hjtauro@tmd.ac.jp (H.T.); s-yoshida.uro@tmd.ac.jp (S.Y.); mntykym.uro@tmd.ac.jp (M.Y.); yoh-m.uro@tmd.ac.jp (Y.M.); y-fujii.uro@tmd.ac.jp (Y.F.); 2Department of Urology, Tsuchiura Kyodo General Hospital, Tsuchiura 300-0028, Ibaraki, Japan; sakai2555@tkgh.jp; 3Department of Urology, Omori Red Cross Hospital, Tokyo 143-8527, Japan; y-ootsuka@omori.jrc.or.jp; 4Department of Urology, Tokyo Metropolitan Ohtsuka Hospital, Tokyo 170-8476, Japan; ryoji_takazawa@tmhp.jp; 5Department of Urology, Tama-Nambu Chiiki Hospital, Tokyo 206-0036, Japan; masatakayano66@gmail.com; 6Department of Urology, Showa General Hospital, Tokyo 187-8510, Japan; tetsuro0215@icloud.com; 7Department of Urology, JA Toride Medical Center, Toride 302-0022, Ibaraki, Japan; kb368u@bma.biglobe.ne.jp; 8Department of Urology, Saitama Red Cross Hospital, Saitama 330-8553, Saitama, Japan; a-slow@plum.plala.or.jp; 9Department of Urology, Kohnodai Hospital, National Center for Global Health and Medicine, Ichikawa 272-8516, Chiba, Japan; nshiho2@ybb.ne.jp; 10Department of Urology, Soka Municipal Hospital, Soka 340-8560, Saitama, Japan; kamata-smh@outlook.com

**Keywords:** diagnosis, hospital mortality, pyelonephritis, sepsis, urinary calculi, urinary tract infections

## Abstract

Prognostic accuracy of the quick sequential organ failure assessment (qSOFA) score for mortality may be limited in elderly patients. Using our multi-institutional database, we classified obstructive acute pyelonephritis (OAPN) patients into young and elderly groups, and evaluated predictive performance of the qSOFA score for in-hospital mortality. qSOFA score ≥ 2 was an independent predictor for in-hospital mortality, as was higher age, and Charlson comorbidity index (CCI) ≥ 2. In young patients, the area under the curve (AUC) of the qSOFA score for in-hospital mortality was 0.85, whereas it was 0.61 in elderly patients. The sensitivity and specificity of qSOFA score ≥ 2 for in-hospital mortality was 80% and 80% in young patients, and 50% and 68% in elderly patients, respectively. For elderly patients, we developed the CCI-incorporated qSOFA score, which showed higher prognostic accuracy compared with the qSOFA score (AUC, 0.66 vs. 0.61, *p* < 0.001). Therefore, the prognostic accuracy of the qSOFA score for in-hospital mortality was high in young OAPN patients, but modest in elderly patients. Although it can work as a screening tool to determine therapeutic management in young patients, for elderly patients, the presence of comorbidities should be considered at the initial assessment.

## 1. Introduction

Sepsis is a life-threatening condition with a higher mortality rate compared with stroke or acute coronary syndrome [1]. Although the definition of sepsis based on a combination of the systemic inflammatory response syndrome (SIRS) score and infection had been used since 1991 [2], the Sepsis-3 Task Force updated its definition as “life threatening organ dysfunction caused by a dysregulated host response to infection” in 2016 [3]. In this new definition, organ dysfunction was defined by the sequential organ failure assessment (SOFA) score, in which six organ systems (hepatic, renal, central nervous, coagulation, cardiovascular, and respiratory) are assessed [3]. For the rapid assessment of patients at high risk of sepsis, the quick SOFA (qSOFA) score, which consists of only three items (respiratory rate ≥ 22 breaths/min, systolic blood pressure ≤ 100 mmHg, and altered mental status), was also proposed as a simpler scoring system [3]. Much attention has been given to the validation of the qSOFA score as an initial screening tool outside the intensive care unit (ICU) since then, and its clinical usefulness was reported [4,5]. Its role as an initial screening tool remains uncertain, however, because several studies reported its low prognostic accuracy [6,7,8].

Obstructive acute pyelonephritis (OAPN) is one of the most commonly encountered urinary tract infections, accounting for 20% to 30% of all septic cases [9,10]. OAPN arises from the obstruction of the urinary tract caused mainly by calculi. Emergent surgeries, such as percutaneous nephrostomy or indwelling ureteral stent, are often required to decompress the urinary tract. If OAPN progresses to severe conditions, such as sepsis or septic shock, the mortality rate can be as high as 10%, and an ICU stay is required [11,12,13]. In our preliminary single-institutional study, the qSOFA score showed good prognostic accuracy for in-hospital mortality in patients with OAPN associated with upper urinary tract calculi [14]. As many as 27% of in-hospital mortalities were missed by the qSOFA score ≥ 2 criteria, however [14]. Given that a substantial number of elderly patients have various comorbidities that can affect their vital signs and mental status, we hypothesized that the prognostic accuracy of the qSOFA score would be compromised in elderly patients. In the current study, we classified OAPN patients into young and elderly groups, and evaluated the prognostic accuracy of qSOFA score for in-hospital mortality using a multi-institutional real-world cohort.

## 2. Materials and Methods

### 2.1. Patients

This retrospective study included 621 patients using a multi-institutional database of OAPN. All patients in this database were admitted and treated for OAPN in 10 Japanese hospitals between March 2010 and October 2018. In patients with multiple admissions for OAPN, we recorded the most severe episode. OAPN with surgical intervention and bilateral OAPN were also included in this study. The diagnosis of OAPN was made on the basis of clinical examination that included fever, flank pain, costovertebral angel tenderness, urinalysis/culture, and blood tests (elevated white blood cell [WBC] count, and serum C-reactive protein [CRP] level) by physicians, and the radiological finding of upper urinary tract obstruction revealed by computed tomography. All patients were treated with antibiotics, with or without emergent decompression of the urinary tract by percutaneous nephrostomy or an indwelling ureteral stent. The drainage method was chosen by the physician. Because there were no available data on respiratory rates, 252 patients were excluded, and the remaining 369 were subject to analysis. The comparison of baseline characteristics between included and excluded patients is shown in Appendix A.

### 2.2. Covariates

We collected the following covariates in the current study: patient age; gender; Charlson comorbidity index (CCI) [15]; comorbidities; body temperature; respiratory rate; heart rate; blood pressure; Glasgow Coma Scale (GCS); urine and blood culture results; obstructive location; drainage method; grade of hydronephrosis; WBC; platelet count; serum creatinine; serum albumin; and serum CRP. The presence of hydronephrosis was evaluated according to the Society for Fatal Urology classification [16]. The physiological and laboratory data measured at initial presentation were used for analysis. The median values were used as cutoffs for age, WBC, platelet count, serum creatinine, serum albumin, and serum CRP.

### 2.3. Calculation of the qSOFA and SIRS Scores

The qSOFA score was calculated by assigning one point for each of respiratory rate ≥ 22 breaths/min, systolic blood pressure ≤ 100 mmHg, and altered mental status. GCS < 15 was regarded as altered mental status [6]. The SIRS score was calculated based on the following four items: temperature > 38.0 °C or < 36.0 °C; heart rate > 90 beats/min; respiratory rate > 20 breaths/min; WBC > 12,000/μL or < 4000/μL; or > 10% immature forms (bands) [7].

### 2.4. Outcomes

The primary and secondary outcomes were in-hospital mortality, and a composite of ICU admission and/or in-hospital mortality, respectively.

### 2.5. Statistical Analysis

The variables were compared between the two groups using the chi-squared test for categorical variables, and the Mann–Whitney U test for continuous variables. The prognostic accuracy of each score was calculated using the area under the curve (AUC) of the receiver operating characteristic (ROC) curve. The optimal cutoff value of each score was determined by the Youden index. Sensitivity, specificity, positive predictive value (PPV), and negative predictive value (NPV) of each score were calculated. Logistic regression analysis was performed to identify the independent predictors of in-hospital mortality and ICU admission. Significant variables in univariate analyses were included in multivariate analyses. Reduced models were constructed by backward elimination. A developed score was internally validated by bootstrap resampling based on 1000 replicates. Statistical analyses were performed using JMP, version 14.0.0 (SAS Institute, Inc., Cary, NC, USA), and R version 4.0.2 (R Foundation for Statistical Computing, Vienna, Austria). Tests with a *p* < 0.05 were considered statistically significant.

## 3. Results

### 3.1. Patient Characteristics

Table 1 shows the patient characteristics according to age. Patients were classified into young (<75 years, *n* = 178) and elderly (≥75 years, *n* = 191) patient groups. The causes of obstruction were calculi, malignancies, and others in 344 (93%), 6 (2%), and 15 (4%) patients, respectively; and 186 (51%) and 118 (33%) patients underwent indwelling ureteral stenting and percutaneous nephrostomy, respectively. While undergoing treatment, 25 (7%) and 70 (19%) patients died in hospital and were admitted to the ICU, respectively. No patients with OAPN associated with malignancies died in hospital during OAPN treatment. Higher age was significantly associated with higher CCI, presence of cardiovascular diseases and stroke, and lower serum albumin (Table 1).

### 3.2. Patient Distribution by the qSOFA Score

A total of 151 (41%), 115 (31%), 75 (20%), and 28 (8%) patients had a qSOFA score of 0, 1, 2, and 3, respectively; and a SIRS score of 0, 1, 2, 3, and 4 was observed in 26 (7%), 69 (19%), 99 (27%), 112 (30%), and 63 (17%) patients, respectively. Distributions of patients who died and were admitted to the ICU by each score and age are shown in Figure 1 and Appendix A, respectively. The qSOFA score was significantly associated with a higher rate of in-hospital mortality in young patients (*p* = 0.014), but not in elderly patients (*p* = 0.11, Figure 1A). The SIRS score was not associated with a higher rate of in-hospital mortality in either age group (Figure 1B). qSOFA score ≥ 3 was significantly associated with in-hospital mortality in elderly patients (*p* = 0.018).

### 3.3. Prognostic Accuracy of the qSOFA Score

For all patients, the prognostic accuracy of the qSOFA score for in-hospital mortality (AUC, 0.68, 95% confidence interval [CI], 0.57–0.79) was significantly higher than that of the SIRS score (AUC, 0.51, 95% CI, 0.41–0.62, *p* = 0.002, Figure 2A). Similarly, the prognostic accuracy of the qSOFA score for the composite of ICU admission and/or in-hospital mortality (AUC, 0.79, 95% CI, 0.74–0.84) was significantly higher than that of the SIRS score (AUC, 0.64, 95% CI, 0.57–0.70, *p* < 0.001, Appendix A). The optimal cutoffs of the qSOFA and SIRS scores were set to ≥ 2 and ≥ 1 according to the Youden index, respectively.

### 3.4. Independent Predictors for In-Hospital Mortality

On multivariate analysis, qSOFA score ≥ 2 (odds ratio [OR], 2.8, *p* = 0.019) was an independent predictor for in-hospital mortality, as was higher age (OR, 2.8, *p* = 0.032), and CCI ≥ 2 (OR, 5.6, *p* = 0.005, Table 2). Nephrostomy or ureteral stent was not significantly associated with in-hospital mortality. qSOFA score ≥ 2 was also independently associated with the composite of ICU admission and/or in-hospital mortality (Appendix A).

### 3.5. Impact of Age on Prognostic Accuracy of the qSOFA Score

In young patients, the AUC of the qSOFA score for in-hospital mortality was 0.85 (95% CI, 0.74–0.96), which significantly outperformed the SIRS score (AUC, 0.55, 95% CI, 0.30–0.80, *p* = 0.023, Figure 2B). In elderly patients, the AUC of the qSOFA score was 0.61 (95% CI, 0.48–0.75), which was statistically similar to that of the SIRS score (0.51, 95% CI, 0.39–0.64, *p* = 0.12, Figure 2C). Analogous findings were obtained for the composite of ICU admission and/or in-hospital mortality; the qSOFA score (AUC, 0.88, 95% CI, 0.83–0.93) had significantly better prognostic accuracy than the SIRS score (AUC, 0.63, 95% CI, 0.53–0.73) in young patients (*p* < 0.001, Appendix A), whereas no significant difference was observed between them in elderly patients (AUC of the qSOFA score, 0.73, 95% CI, 0.65–0.80, AUC of the SIRS score, 0.65, 95% CI, 0.57–0.74, *p* = 0.10, Appendix A).

Table 3 shows the sensitivity, specificity, PPV, and NPV of the qSOFA and SIRS scores for in-hospital mortality according to age. The sensitivity, specificity, PPV, and NPV of qSOFA score ≥ 2 for in-hospital mortality were 80%, 80%, 10%, and 99% in young patients, and 50%, 68%, 16%, and 92% in elderly patients, respectively.

### 3.6. Development of the CCI-Incorporated qSOFA Score for Elderly Patients

In the logistic analysis for elderly patients, CCI ≥ 2 (OR, 8.1, *p* = 0.008) and qSOFA score ≥ 3 (OR, 3.6, *p* = 0.047) were independent predictors for in-hospital mortality (Appendix A). qSOFA score ≥ 2 was not significantly associated with in-hospital mortality. As such, we developed the CCI-incorporated qSOFA score by assigning one point for CCI ≥ 2, and then added it to the qSOFA score. In elderly patients, the incorporation of the CCI into the qSOFA score significantly improved its AUC from 0.61 to 0.66 (95% CI, 0.53–0.79, *p* < 0.001, Figure 2D). After the internal validation by bootstrap resampling, the AUC of CCI-incorporated qSOFA score was 0.66. The optimal cutoff of CCI-incorporated qSOFA score was ≥3 according to the Youden index. The sensitivity, specificity, PPV, and NPV of CCI-incorporated qSOFA ≥ 3 were 50%, 74%, 18%, and 93%, respectively (Table 3).

## 4. Discussion

The current study demonstrated the different predictive performance of the qSOFA score for in-hospital mortality between young and elderly OAPN patients; the qSOFA score predicted in-hospital mortality with high accuracy in young patients (AUC, 0.85), whereas its prognostic accuracy was modest in elderly patients (AUC, 0.61). This trend was also observed in several previous studies, in which the AUC of the qSOFA score was less than 0.70 in elderly patients [17,18,19]. The sensitivity and specificity of qSOFA score ≥ 2 for in-hospital mortality were 80% and 80% in young patients, and 50% and 68% in elderly patients, respectively. Analogous results were obtained for the composite of ICU admission and/or in-hospital mortality. Thus, in young patients, qSOFA score ≥ 2 can serve as an initial screening tool to determine intensive therapeutic plans, such as ICU care and emergent decompression of the urinary tract in the acute phase management of OAPN. It may not be sufficiently sensitive as a screening tool for elderly patients, however.

We developed the CCI-incorporated qSOFA score for elderly patients. Several previous studies have reported the combined qSOFA score and CCI, and its improved prognostic accuracy [20,21]. Given the high prognostic accuracy of the qSOFA score in young patients, the use of the CCI-incorporated qSOFA score may be limited to elderly patients. Intensive treatment is recommended for elderly patients with CCI-incorporated qSOFA ≥ 3 in clinical practice.

In elderly patients, the criteria of qSOFA score of ≥ 2 missed 10 deceased patients in the current study. Meanwhile, only one deceased patient was missed by qSOFA score of ≥ 2 in young patients. This suggests that elderly patients should be intensively monitored after admission, even if their qSOFA scores at initial presentation are ≤ 1. Our previous study showed that SOFA score ≥ 7 could identify high-risk patients with low qSOFA scores [14]. Because the SOFA score requires a number of laboratory tests [1,22], it can be useful in the further identification of patients at high risk of sepsis and mortality, especially in elderly patients, after the initial screening by qSOFA score and CCI.

Why is the prognostic accuracy of the qSOFA score lower in elderly patients? One possibility is that some elderly patients may have died of other diseases as a result of the severe course of OAPN, as they tended to have more comorbidities. Indeed, of the elderly patients in our cohort, three (15%) and six (30%) patients died of other comorbidities and pneumonia, respectively, whereas all deceased young patients died of OAPN (Appendix A). Second, the qSOFA score is likely to be overestimated in elderly patients because their comorbidities can increase their scores. Third, elderly patients might not have been treated properly because they generally tend to receive conservative treatments [23,24]. However, higher age was not significantly associated with a lower rate of drainage in the current study. Fourth, OAPN might have progressed, even though their disease severity was mild at initial presentation. This indicates that continuous evaluation using the SOFA score can be important in the management of elderly OAPN patients.

The current study has several limitations to be addressed. First, selection biases may have occurred due to the retrospective nature of the current study. Our findings should be validated by prospective studies. Second, 41% of patients were excluded from the analysis due to missing values in respiratory rates. Excluded patients tended to have a less severe disease course compared with included patients. Third, the current study did not evaluate the SOFA score because it was not available in our multi-institutional database. However, the aim of the current study is to assess the prognostic accuracy of the qSOFA score as a screening tool. Because the SOFA score is not suitable as a screening tool, it should be used after evaluation of the qSOFA score. Finally, in the current study, the rate of in-hospital mortality was 7%, which appears to be relatively high. This is, however, nearly equivalent to the rate in previous studies [25,26].

## 5. Conclusions

The qSOFA score predicted in-hospital mortality with high accuracy in young OAPN patients, whereas its prognostic accuracy was modest in elderly OAPN patients. Thus, in young patients, the qSOFA score can be considered a good initial screening tool to determine the acute phase management of OAPN. For elderly patients, however, the presence of comorbidities should be considered at the initial assessment.

## Figures and Tables

**Figure 1 diagnostics-11-02277-f001:**
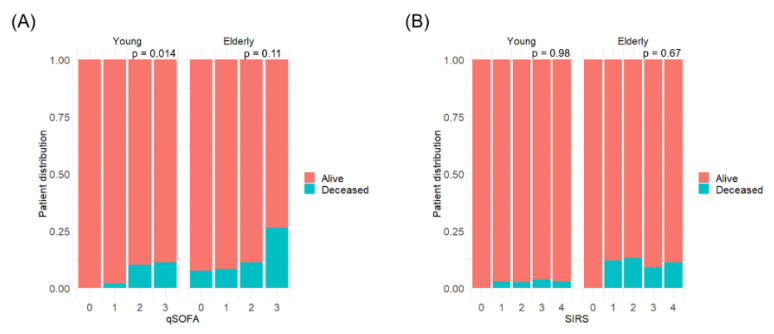
Distribution of deceased patients by qSOFA and SIRS scores. (**A**), Distribution by qSOFA score in young (left) and elderly (right) patients. (**B**), Distribution by SIRS score in young (left) and elderly (right) patients.

**Figure 2 diagnostics-11-02277-f002:**
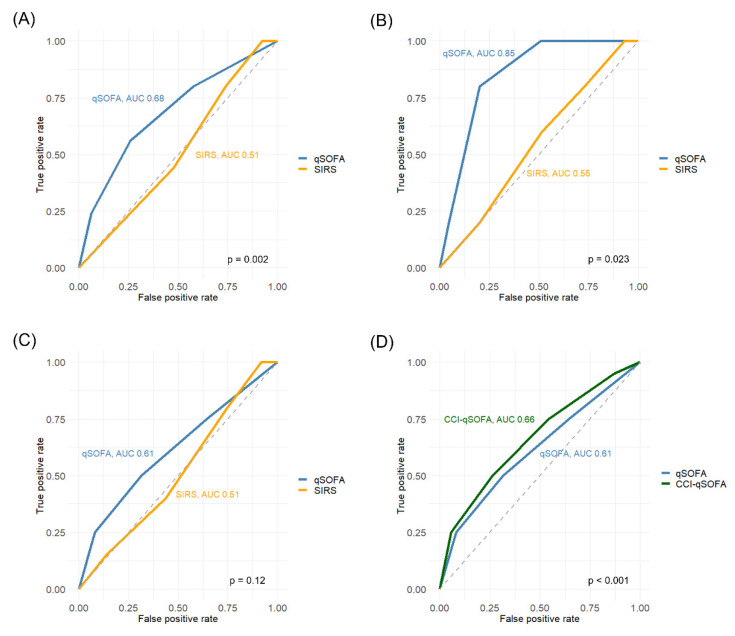
ROC curves of the qSOFA, SIRS, and CCI-incorporated qSOFA scores in predicting in-hospital mortality. (**A**), ROC curves of the qSOFA and SIRS scores in predicting in-hospital mortality in total patients, (**B**), ROC curves of the qSOFA and SIRS scores in predicting in-hospital mortality in young patients, (**C**), ROC curves of the qSOFA and SIRS scores in predicting in-hospital mortality in elderly patients, (**D**), ROC curves of the qSOFA and CCI-incorporated qSOFA scores in predicting in-hospital mortality in elderly patients.

**Table 1 diagnostics-11-02277-t001:** Patient characteristics.

Variables		Total, n (%)	Young, n (%)	Elderly, n (%)	*p* Value
No. of patients		369 (100)	178 (48)	191 (52)	
Sex	Females	258 (70)	130 (73)	128 (67)	0.21
	Males	111 (30)	48 (27)	63 (33)	
CCI	0–1	145 (39)	92 (52)	53 (28)	<0.001
	≥2	224 (61)	86 (48)	138 (72)	
DM	Yes	76 (21)	36 (20)	40 (21)	0.86
	No	293 (79)	142 (80)	151 (79)	
CVD	Yes	106 (29)	40 (22)	66 (35)	0.01
	No	263 (71)	138 (78)	125 (65)	
Stroke	Yes	65 (18)	19 (11)	46 (24)	<0.001
	No	304 (82)	159 (89)	145 (76)	
Malignancy	Yes	71 (19)	29 (16)	42 (22)	0.17
	No	298 (81)	149(84)	149 (78)	
Immunosuppression	Yes	26 (7)	18 (10)	8 (4)	0.026
	No	343 (93)	160 (90)	183 (96)	
WBC (103/μL), median (range)		12.4 (0.800–61.5)	12.6 (1.2–61.5)	12 (0.8–39.2)	0.41
Platelet (104/μL), median (range)		15.7 (0.2–57.0)	16.6 (0.2–53.8)	14.9 (1.3–57)	0.088
CRP (mg/dL), median (range)		14.3 (0.0–52.1)	14.1 (0–52.1)	14.5 (0.1–43.6)	0.45
Albumin (g/dL), median (range)		3.1 (1.5–4.9)	3.3 (1.7–4.9)	2.9 (1.5–4.7)	<0.001
Creatinine (mg/dL), median (range)		1.2 (0.3–8.0)	1.2 (0.3–8.0)	1.2 (0.3–8.0)	0.87
Midstream urine culture	Positive	279 (76)	137 (77)	141 (74)	0.48
	Negative	80 (22)	38 (21)	42 (22)	
	None	10 (3)	3 (2)	7 (4)	
Blood culture	Positive	143 (39)	55 (31)	88 (46)	0.010
	Negative	160 (43)	89 (50)	71 (37)	
	None	66 (18)	34 (19)	32 (17)	
Position of obstruction	Renal calyx or pelvis	9 (2)	3 (2)	6 (3)	0.33
	Pelvic ureteral junction	53 (14)	30 (17)	23 (12)	
	Upper ureter	168 (45)	78 (44)	90 (47)	
	Mid ureter	47 (13)	27 (15)	20 (10)	
	Lower ureter	82 (22)	37 (21)	45 (24)	
	Unknown	10 (3)	3 (2)	7 (4)	
Laterality	Right	165 (45)	81 (46)	84 (44)	0.049
	Left	194 (53)	96 (54)	98 (51)	
	Bilateral	10 (3)	1 (1)	9 (5)	
Hydronephrosis	Low grade (0–2)	219 (59)	116 (65)	103 (54)	0.026
	High grade (3–4)	146 (40)	60 (34)	86 (45)	
	Unknown	4 (1)	2 (1)	2 (1)	
Cause of obstruction	Calculus	344 (93)	167 (94)	177 (93)	0.93
	Tumor	6 (2)	3 (1.7)	3 (2)	
	Others	15 (4)	6 (3)	9 (5)	
	Unknown	4 (1)	2 (1)	2 (1)	
Method of drainage	Nephrostomy	118 (32)	60 (34)	58 (31)	0.39
	Ureteral stent	186 (50)	93 (53)	93 (50)	
	None	65 (18)	24 (14)	35 (19)	
DIC	Yes	93 (25)	43 (24)	50 (26)	0.66
	No	276 (75)	135 (76)	141 (74)	
ICU admission	Yes	70 (19)	29 (16)	41 (21)	0.21
	No	299 (81)	149 (84)	150 (79)	
In-hospital mortality	Yes	25 (7)	5 (3)	20 (10)	0.003
	No	344 (93)	173 (97)	171 (90)	

Values are presented as number (%) or median (range). CCI, Charlson comorbidity index; CRP, C-reactive protein; CVD, cerebrovascular disease; DIC, disseminated intravascular coagulation; DM, diabetes mellitus; ECOG-PS, Eastern Cooperative Oncology Group performance status; qSOFA, quick sequential organ failure assessment; SIRS, systemic inflammatory response syndrome; WBC, white blood cell.

**Table 2 diagnostics-11-02277-t002:** Logistic regression analysis for in-hospital mortality.

Variables		Univariate	Multivariate		
		*p* Value	OR	95%CI	*p* Value
Sex	Males vs. females	0.81			
Age	Elderly vs. young	0.002	2.8	1.1–8.8	0.032
CCI	≥2 vs. 0–1	<0.001	5.6	1.6–35.7	0.005
DM	Yes vs. No	0.67			
CVD	Yes vs. No	0.21			
Stroke	Yes vs. No	0.022			
Malignancy	Yes vs. No	0.32			
Immunosuppression	Yes vs. No	0.36			
Cause of obstruction	Calculus vs. Others	0.81			
	Tumor vs. Others	0.36			
Laterality	Right vs. Left	0.62			
Hydronephrosis	Low grade (0–2) vs. High grade (3–5)	0.55			
CRP	≥14.3 vs. <14.3	0.54			
Platelet	<15.7 vs. ≥15.7	0.058			
WBC	≥12.4 vs. <12.4	0.54			
Cre	≥1.2 vs. <1.2	0.060			
Drainage	Nephrostomy vs. None	0.49			
	Ureteral stent vs. None	0.97			
SIRS	≥1 vs. 0	0.035			
qSOFA	≥2 vs. 0–1	0.002	2.8	1.2–6.6	0.019

CCI, Charlson comorbidity index; CI, confidence interval; CRP, C-reactive protein; CVD, cerebrovascular disease; DM, diabetes mellitus; OR, odds ratio; qSOFA, quick sequential organ failure assessment; SIRS, systemic inflammatory response syndrome; WBC, white blood cell.

**Table 3 diagnostics-11-02277-t003:** Sensitivity, specificity, PPV, and NPV of the qSOFA, SIRS, and CCI-incorporated qSOFA scores for in-hospital mortality.

Total Cohort					
		Sensitivity	Specificity	PPV	NPV
qSOFA	≥1	80%	42%	9%	97%
	≥2	56%	74%	14%	96%
	≥3	24%	94%	21%	94%
SIRS	≥1	100%	8%	7%	100%
	≥2	80%	26%	7%	95%
	≥3	44%	52%	6%	93%
	≥4	16%	83%	6%	93%
Young patients					
		Sensitivity	Specificity	PPV	NPV
qSOFA	≥1	100%	49%	5%	100%
	≥2	80%	80%	10%	99%
	≥3	20%	95%	11%	98%
SIRS	≥1	100%	7%	3%	100%
	≥2	80%	27%	3%	98%
	≥3	60%	49%	3%	98%
	≥4	20%	80%	3%	97%
Elderly patients					
		Sensitivity	Specificity	PPV	NPV
qSOFA	≥1	75%	36%	12%	92%
	≥2	50%	68%	16%	92%
	≥3	25%	92%	26%	91%
SIRS	≥1	100%	8%	11%	100%
	≥2	80%	25%	11%	91%
	≥3	40%	56%	10%	89%
	≥4	15%	86%	11%	90%
CCI-qSOFA	≥1	95%	13%	11%	96%
	≥2	75%	46%	14%	94%
	≥3	50%	74%	18%	93%
	≥4	25%	94%	33%	91%

CCI, Charlson Comorbidity Index; ICU, intensive care unit; NPV, negative predictive value; PPV, positive predictive value; qSOFA, quick sequential organ failure assessment; SIRS, systemic inflammatory response syndrome.

## Data Availability

Data available on request due to privacy/ethical restrictions.

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
