# Peer review of "Prognostic Accuracy of the qSOFA Score for In-Hospital Mortality in Elderly Patients with Obstructive Acute Pyelonephritis: A Multi-Institutional Study"

_diagnostics, 2021, doi:10.3390/diagnostics11122277_

Round 1

Reviewer 1 Report

In this study, the authors used a multi-institutional database of hospitalized patients with acute obstructive pyelonephritis to validate the prognostic accuracy of the qSOFA and SIRS scores in the young and the elderly for in-hospital mortality. They also developed a CCI-qSOFA score.

This is an interesting manuscript. However, I have a few major concerns:
1) Selection bias - The authors excluded 252/621 (41%) due to missing respiratory rates. This is a major limitation. More so without a thorough examination of the missingness pattern and comparing the missing patients' characteristics to those with complete data. Possibly, the 369 included patients are not representative of the population of interest.
2) Overfitting - The manuscript includes two different study designs: a validation study (qSOFA, SIRS) and the development of a new score (CCI-qSOFA). The new score requires either internal validation (a test set), external validation (in an institution not included in the development), or preferably both. Otherwise, overfitting and optimistic results are highly likely.
3) Competing Risks - setting the secondary outcome as ICU admission alone ignores the competing risk of in-hospital mortality. If this secondary outcome is to imply patient severity, a composite outcome of ICU admissions and death would be more accurate.
4) Problematic measure of model utility - AUC is not very informative for clinical decision-making. An emphasis on sensitivity, specificity, NPV, and PPV instead would significantly improve clinical understanding.

Author Response

Dear the Reviewer,

We greatly appreciated the reviewer for taking their precious time to review our paper and giving constructive comments. We revised the manuscript according to the reviewers’ comments. Please take a look at the changes we made in the revised version outlined below.

In this study, the authors used a multi-institutional database of hospitalized patients with acute obstructive pyelonephritis to validate the prognostic accuracy of the qSOFA and SIRS scores in the young and the elderly for in-hospital mortality. They also developed a CCI-qSOFA score.
This is an interesting manuscript. However, I have a few major concerns:
1) Selection bias - The authors excluded 252/621 (41%) due to missing respiratory rates. This is a major limitation. More so without a thorough examination of the missingness pattern and comparing the missing patients' characteristics to those with complete data. Possibly, the 369 included patients are not representative of the population of interest.

We appreciate the reviewer’s comment. As the reviewer is pointing out, this issue is a major limitation of this study. According the comparison of baseline characteristics between included and excluded patients, excluded patients tended to have a less severe disease course, suggesting that measurement of respiratory rates might have been clinically unnecessary in the management of them.

Thus, we added Table S1, which shows the comparison of baseline characteristics between included and excluded patients. Moreover, we added the following sentence to the limitation subsection of the Discussion section: “Second, 41% of patients were excluded from analysis due to missing values in respiratory rates. Excluded patients tended to have a less severe disease course compared with included patients.” (line 268-270)

2) Overfitting - The manuscript includes two different study designs: a validation study (qSOFA, SIRS) and the development of a new score (CCI-qSOFA). The new score requires either internal validation (a test set), external validation (in an institution not included in the development), or preferably both. Otherwise, overfitting and optimistic results are highly likely.

We appreciate the reviewer’s constructive comment. We performed the internal validation using bootstrap resampling with 1,000 replicates.

Thus, we added the following sentences to the manuscript:

“A developed score was internally validated by bootstrap resampling based on 1,000 replicates.” (line 115-117)

“After the internal validation by bootstrap resampling, the AUC of CCI-incorporated qSOFA score was 0.66.” (line 223-224)

3) Competing Risks - setting the secondary outcome as ICU admission alone ignores the competing risk of in-hospital mortality. If this secondary outcome is to imply patient severity, a composite outcome of ICU admissions and death would be more accurate.

We appreciate the reviewer’s constructive comment. We changed the secondary outcome into a composite of ICU admission and/or in-hospital mortality. The results are shown in Figure S2 and Table S2.

Moreover, we revised the following sentences:

“The primary and secondary outcomes were in-hospital mortality and a composite of ICU admission and/or in-hospital mortality, respectively.” (line 104-105)

“Similarly, the prognostic accuracy of the qSOFA score for the composite of ICU admission and/or in-hospital mortality (AUC, 0.79, 95% CI, 0.74–0.84) was significantly higher than that of the SIRS score (AUC, 0.64, 95% CI, 0.57–0.70, p < 0.001, Figure S2A).” (line 154-157)

“qSOFA score ≥ 2 was also independently associated with the composite of ICU admission and/or in-hospital mortality (Table S2).” (line 171-173)

“Analogous findings were obtained for the composite of ICU admission and/or in-hospital mortality; the qSOFA score (AUC, 0.88, 95% CI, 0.83–0.93) had significantly better prognostic accuracy than the SIRS score (AUC, 0.63, 95% CI, 0.53–0.73) in young patients (p < 0.001, Figure S2B) whereas no significant difference was observed between them in elderly patients (AUC of the qSOFA score, 0.73, 95% CI, 0.65–0.80, AUC of the SIRS score, 0.65, 95% CI, 0.57–0.74, p = 0.10, Figure S2C).” (line 189-195)

4) Problematic measure of model utility - AUC is not very informative for clinical decision-making. An emphasis on sensitivity, specificity, NPV, and PPV instead would significantly improve clinical understanding.

We appreciated the reviewer’s pointing out. We summarized the results of sensitivity, specificity, PPV, NPV of the qSOFA, SIRS, and CCI-qSOFA scores in Table 3.

Reviewer 2 Report

This manuscript retrospectively investigated the relationship between qSOFA and obstructive pyelonephritis.

Minor comments

Both qSOFA and SOFA scores are common to emergency physicians, but not common to general urologists, so it is better to give a more general explanation of the scores in the introduction.

Elderly people are over 75 years old in this study, was there any difference if they were separated by 65 or 70 years old? And what if you have 65 to 75 years old as one of the section?

In the elderly, there seems to be a difference between survivors and dead at qSOFA scores of 2 or less and 3 points. Is there a significant difference between them?

Was there a relationship between death and the patients that were conservatively followed?

Author Response

Dear the Reviewer,

We greatly appreciated the reviewer for taking their precious time to review our paper and giving constructive comments. We revised the manuscript according to the reviewers’ comments. Please take a look at the changes we made in the revised version outlined below.

This manuscript retrospectively investigated the relationship between qSOFA and obstructive pyelonephritis.

Minor comments

Both qSOFA and SOFA scores are common to emergency physicians, but not common to general urologists, so it is better to give a more general explanation of the scores in the introduction.

We appreciate the reviewer’s constructive comment. We revised the Introduction section as follows: “In this new definition, organ dysfunction was defined by the Sequential Organ Failure Assessment (SOFA) score, in which six organ systems (hepatic, renal, central nervous, coagulation, cardiovascular, and respiratory) are assessed [3]. For the rapid assessment of patients at high risk of sepsis, the quick SOFA (qSOFA) score, which consists of only three items (respiratory rate ≥ 22 breaths/min, systolic blood pressure ≤ 100 mmHg, and altered mental status), was also proposed as a simpler scoring system [3].” (line 46-51)

Elderly people are over 75 years old in this study, was there any difference if they were separated by 65 or 70 years old? And what if you have 65 to 75 years old as one of the section?

We appreciate the reviewer’s suggestion. We calculated the AUC of the qSOFA score using various cutoff points. The AUC was 0.84 in patients < 65 years and 0.65 in patients ≥ 65 years, 0.83 in patients < 70 years and 0.64 in patients ≥ 70 years, and 0.85 in patients < 75 years and 0.61 in patients ≥ 75 years. The AUC was mostly separately using the cutoff of 75 years, and thus we considered 75 years as the best cutoff and used it in this study.

Moreover, the AUC of the qSOFA score was 0.83 in patients with 65-75 years, which is comparable to that in patients < 65 years. Thus, we did not add the cutoff of 65 years in this study.

In the elderly, there seems to be a difference between survivors and dead at qSOFA scores of 2 or less and 3 points. Is there a significant difference between them?

We appreciated the reviewer’s comment. As the reviewer is pointing out, qSOFA ≥ 3 was significantly associated with in-hospital mortality in elderly patients (p = 0.018). Thus, we added the following sentence to the Results section: “qSOFA score ≥ 3 was significantly associated with in-hospital mortality in elderly patients (p = 0.018)” (line 139-140)

Moreover, qSOFA ≥ 3 vs. 0-2 was added as a variable in the logistic regression analysis for in-hospital mortality in elderly patients (Table S3) and qSOFA ≥ 3 (OR, 3.6, p = 0.047) was an independent predictor for in-hospital mortality along with CCI ≥ 2 (OR, 8.1, p = 0.008). Thus, we revised the following sentences in the Results section: “In the logistic analysis for elderly patients, CCI ≥ 2 (OR, 8.1, p = 0.008) and qSOFA score ≥ 3 (OR, 3.6, p = 0.047) were independent predictors for in-hospital mortality (Table S3). qSOFA score ≥ 2 was not significantly associated with in-hospital mortality.” (line 217-220)

Was there a relationship between death and the patients that were conservatively followed?

We appreciate the reviewer’s comment. Nephrostomy or ureteral stent, which was not performed in patients who were conservatively followed, was not significantly associated with in-hospital mortality in this study (Table 2). We added the following sentence to the Results section: “Nephrostomy or ureteral stent was not significantly associated with in-hospital mortality.” (line 170-171)

Round 2

Reviewer 1 Report

I applaud the authors for their work in addressing the reviewers' comments and improving this manuscript.

Two issues:

1) Do the authors recommend using the developed CCI-qSOFA score in the elderly (if so, with what cutoff?) or considering the presence of comorbidities individually? It is unclear from the discussion.

2) The resulting sensitivities, specificities, NPVs, PPVs of the CCI-qSOFA are presented in Table 3. Still, the authors may wish to emphasize it in the results, at least for the recommended threshold, and in the discussion to encourage/discourage the use of the developed score. 

Author Response

Dear the Reviewer,

We greatly appreciated the reviewer for taking their precious time to review our paper again and giving constructive comments. We revised the manuscript according to the reviewer’ comments. Please take a look at the changes we made in the revised version outlined below.

I applaud the authors for their work in addressing the reviewers' comments and improving this manuscript.

Two issues:

1) Do the authors recommend using the developed CCI-qSOFA score in the elderly (if so, with what cutoff?) or considering the presence of comorbidities individually? It is unclear from the discussion.

We appreciate the reviewer’s pointing out. We recommend the use of CCI-qSOFA for elderly patients. The optimal cutoff of CCI-qSOFA was ≥ 3 according to the Youden index. Thus, we added the following sentence to the Results section:

“The optimal cutoff of CCI-incorporated qSOFA score was ≥ 3 according to the Youden index.” (line 219-220)

Moreover, we revised the Discussion section as follows:

“Intensive treatment is recommended for elderly patients with CCI-incorporated qSOFA ≥ 3 in clinical practice.” (line 239-240)

2) The resulting sensitivities, specificities, NPVs, PPVs of the CCI-qSOFA are presented in Table 3. Still, the authors may wish to emphasize it in the results, at least for the recommended threshold, and in the discussion to encourage/discourage the use of the developed score. 

We appreciate the reviewer’s constructive comment. We described the sensitivity, specificity, PPV, and NPV of CCI-qSOFA ≥ 3 in the Results section and added the comment to encourage its use for elderly patients in the Discussion section as follows:

“The sensitivity, specificity, PPV, and NPV of CCI-incorporated qSOFA ≥ 3 were 50%, 74%, 18%, and 93%, respectively (Table 3).” (line 220-221)

“Intensive treatment is recommended for elderly patients with CCI-incorporated qSOFA ≥ 3 in clinical practice.” (line 239-240)